# TRIPLE-S: A STICKER SEMANTIC SIMILARITY BENCHMARK WITH GENERAL STICKER ENCODER

## ABSTRACT

Stickers have become a popular form of visual communication, yet understanding their semantic relationships remains challenging due to their highly diverse and symbolic content. In this work, we formally **define the Sticker Semantic Similarity task** and introduce **Triple-S**, the first benchmark for this task, consisting of 905 human-annotated positive and negative sticker pairs. Through extensive evaluation, we show that existing pretrained vision and multimodal models struggle to capture nuanced sticker semantics. To address this, we propose the **General Sticker Encoder (GSE)**, a lightweight and versatile model that learns robust sticker embeddings using both Triple-S and additional datasets. GSE achieves superior performance on unseen stickers, and demonstrates strong results on downstream tasks such as emotion classification and sticker-to-sticker retrieval. By releasing both Triple-S and GSE, we provide standardized evaluation tools and robust embeddings, enabling future research in sticker understanding, retrieval, and multimodal content generation. The Triple-S benchmark and GSE have been publicly released and are available here [1].

## 1 INTRODUCTION

Stickers are ubiquitous in online communication, serving as a rich and expressive medium for conveying emotions and ideas Chee et al. (2025b). Beyond casual use, recent research has begun to explore personalized sticker generation Xu et al. (2024); Shen et al. (2024) with semantic preservation Xu et al. (2025). However, progress in this area is limited by the absence of a standardized sticker semantic similarity evaluation benchmark. Current works resort to general-purpose vision encoders such as CLIP Radford et al. (2021), DINOv2 Oquab et al. (2023), or ViT Dosovitskiy et al. (2021) to approximate sticker semantic similarity. Yet, the applicability of these encoders to sticker semantics has never been systematically validated, leaving a critical gap in evaluation.

Developing a benchmark for sticker semantic similarity presents us with a main challenge. *Benchmark dataset construction* is inherently difficult. While coarse sticker classification datasets exist, a high-quality evaluation requires carefully curated positive and hard negative pairs. Human annotation of sticker semantics is both subjective and labor-intensive, making the creation of such a dataset tedious and costly. In addition to existing baselines, providing a feasible *evaluation method* pose challenges. Though, general purpose vision-language models are capable of natural language, possibly sticker semantics Achiam et al. (2024), they are computationally intensive and impractical for large-scale or rapid evaluation. This motivates the need for a lightweight, efficient, and versatile encoder tailored to the unique properties of stickers for the sticker semantic similarity task.

We formally define the **sticker semantic similarity** task, which evaluates whether two stickers convey similar meaning, Figure 1. To standardize evaluation, we introduce **Triple-S**, the first benchmark for this task, consisting of 905 human-annotated sticker pairs labeled as positive or negative with agreement from annotators of different backgrounds. Then, we benchmark several general-purpose image encoders such as CLIP, DINOv2, and ViT on Triple-S, highlighting their limitations. Finally, we present **GSE (General Sticker Encoder)**, a lightweight and effective baseline that surpasses existing methods on unseen data and demonstrates transfer to downstream tasks such as emotion classification and top-$k$ image-to-image retrieval.

---

[1] https://anonymous.4open.science/r/triple-s-6E65/

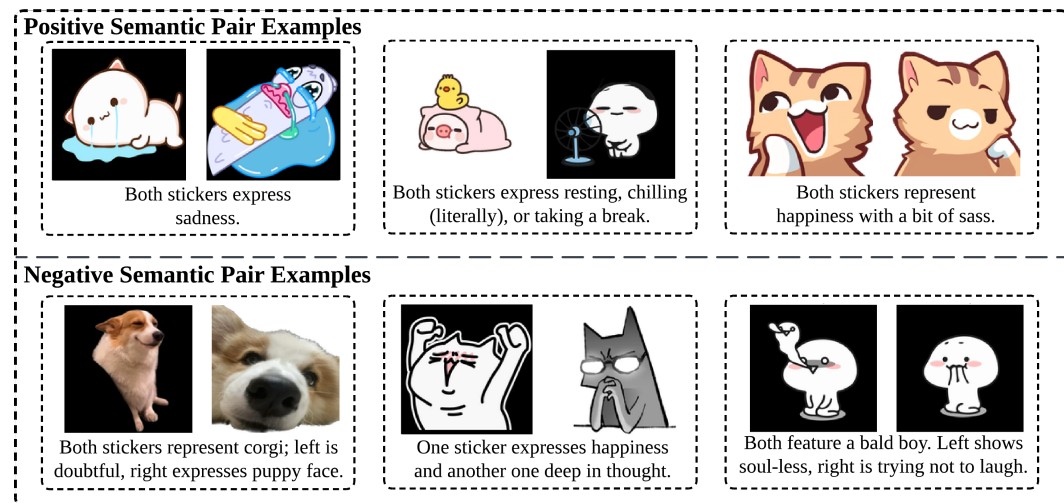

Figure 1: Examples of semantic pairings. The top row shows positive pairs, where both stickers convey similar emotions or actions. The bottom row shows negative pairs, where stickers differ in emotion, expression, or context despite visual or thematic similarities.

We summarize our contributions:

First, we propose and formally define the Sticker Semantic Similarity task.

Second, we introduce **Triple-S**, the first benchmark for the visual sticker semantic similarity task, consisting of a curated dataset of 905 annotated semantically positive and negative sticker pairs.

Third, we benchmark common strong pretrained vision encoders on Triple-S benchmark, revealing their limitations in capturing sticker semantics.

Finally, we present **GSE (General Sticker Encoder)**, a lightweight and versatile sticker encoder that provides a strong baseline for the 3S task and generalizes effectively to other related tasks.

## 2 RELATED WORK

### 2.1 STICKER SEMANTIC SIMILARITY EVALUATION

**Image Encoders**   Recent vision-language models, such as CLIP Radford et al. (2021), ViT Dosovitskiy et al. (2021), and DINOv2 Oquab et al. (2023), provide powerful image embeddings for capturing semantic content. These models have been successfully applied to various tasks, including image retrieval and similarity evaluation. However, they are not ideally suited for stickers. Stickers often convey meaning through stylized, exaggerated, or culturally specific visual cues that general-purpose models fail to capture. While StickerCLIP Zhao et al. (2023) attempts to address this, it is closed-source and not publicly accessible, limiting its use for research and evaluation.

**Vision-Language Model**   Recent large vision-language models (VLMs), including OpenAI's ChatGPT series Achiam et al. (2024) and ChatGLM Zeng et al. (2023) variants with visual input, have achieved remarkable progress in understanding complex image-text relationships. They enable zero-shot classification, retrieval, and semantic similarity evaluation by jointly modeling visual and textual information. Despite their power, these models are not ideally suited for stickers. They can hallucinate or misinterpret stylized, exaggerated, or culturally specific visual cues, which are common in sticker designs. Moreover, they are resource-intensive and computationally expensive, making them impractical for large-scale or real-time evaluation tasks where lightweight and efficient metrics are required.

## 2.2 STICKER DATASETS AND BENCHMARKS

With the rise of instant messaging, sticker datasets have grown significantly, but given the diversity and complexity of sticker tasks, existing datasets still represent only the tip of the iceberg. We broadly categorize these datasets into five groups: dialogues, sentiment, captions, multimedia, and retrieval.

Dialogue datasets typically contain truncated conversations with stickers Gao et al. (2021; 2020); Fei et al. (2021); Liang et al. (2024); Shi & Kong (2024); Zhang (2024); Wang et al. (2025). Notably, U-Sticker Chee et al. (2025a) provides continuous user information, temporal context, and domain-specific sticker conversations. While some datasets are publicly available, others remain private or inaccessible. Sentiment datasets focus on predicting the sentiment conveyed by stickers, either through binary labels Ge et al. (2022) or fine-grained classifications Liu et al. (2022). Certain datasets, such as Sticker820K Zhao et al. (2023), also include OCR text or descriptive metadata. Among these, SER30K Liu et al. (2022) is publicly available. Caption datasets contain stickers with textual elements, e.g., TGIF Li et al. (2016) provides GIF captions. Unfortunately, StickerTag Wang et al. (2024b) is unavailable for analysis. Multimedia datasets provide raw sticker images for general use or generation tasks. ChineseB2B Zhaoolee (2024) offers a large collection of images, while VSD2M Yuan et al. (2025) includes an extensive repository of sticker animations. Retrieval datasets pair user search queries with corresponding sticker images, capturing underlying sticker usage intent. StickerQueries provides bilingual (English and Chinese) queries curated by native speakers. PerSRV Chee et al. (2024), another sticker–query pair dataset, is not publicly available.

However, as can be seen in Table 1, none of the existing datasets are suitable for evaluating sticker semantic similarity. Sentiment datasets are too coarse to capture fine-grained semantics, while retrieval-based datasets like StickerQueries are unreliable for distinguishing subtle nuances such as irony, sarcasm, or implied meaning. Consequently, a human-annotated, pairwise dataset remains essential for the sticker semantic similarity task.

Table 1: Sticker datasets summary, highlighting the need for fine-grained semantic similarity benchmarks. Analyses dimensions include human annotation (**Human.**), semantic pair construction capability (**Pair-able**), granularity level (**Pair Quality**)—ranging from fine (e.g., *"sobbing" vs. "bawling eyes"*), medium (e.g., *"thank you" vs. "thank you" in different styles*), to coarse (e.g., *"sad" vs. "sad"*). Additional factors include encoder provision and public access. Triple-S provides fine-grained pairs with both benchmark and general encoder.

| Dataset | Human.? | Pair-able? | Pair Quality? | Has Gen. Encoder? | Pub. Avail? |
|---|---|---|---|---|---|
| *Multimedia* | | | | | |
| VSD2M Yuan et al. (2025) | ✗ | ✗ | N/A | ✗ | ✓ |
| ChineseB2B Zhaoolee (2024) | ✗ | ✗ | N/A | ✗ | ✓ |
| *Dialogues* | | | | | |
| SRS, PERSRS Gao et al. (2020) | ✗ | ✗ | N/A | ✗ | ✗ |
| MOD Fei et al. (2021) | ✗ | ✗ | N/A | ✗ | ✓ |
| MCDSCS Shi & Kong (2024) | ✓ | ✗ | N/A | ✗ | ✓ |
| STICKERCONV Zhang (2024) | ✗ | ✗ | N/A | ✗ | ✓ |
| U-Sticker Chee et al. (2025a) | ✗ | ✗ | ✗ | ✗ | ✓ |
| MultiChat Wang et al. (2025) | ✓ | ✓ | Medium | ✗ | ✓ |
| *Sentiment Classification* | | | | | |
| SER30K Liu et al. (2022) | ✓ | ✓ | Coarse | ✗ | ✓ |
| CSMSA Ge et al. (2022) | N/A | N/A | N/A | ✗ | ✗ |
| Sticker820K Zhao et al. (2023) | N/A | N/A | N/A | ✓ | ✗ |
| *Captions* | | | | | |
| TGIF Li et al. (2016) | ✓ | ✗ | N/A | ✗ | ✓ |
| MET-Meme Liaolian (2023) | ✓ | ✓ | Coarse | ✗ | ✓ |
| StickerTag Wang et al. (2024b) | N/A | N/A | N/A | ✗ | ✗ |
| *Retrieval* | | | | | |
| WXChallenge WeChat (2023) | ✓ | ✓ | Coarse | ✗ | ✗ |
| StickerQueries Chee et al. (2025b) | ✓ | ✓ | Coarse | ✗ | ✓ |
| *Visual Semantic Similarity* | | | | | |
| **Triple-S (Ours)** | ✓ | ✓ | **Fine** | ✓ | ✓ |

## 3 TRIPLE-S BENCHMARK

### 3.1 STICKER SEMANTIC SIMILARITY TASK

We formally propose the *visual sticker semantic similarity task* where the main objective is to predict whether two stickers, represented solely by their pixel data $s_i, s_j \in \mathcal{X}$, are semantically similar ($y_{ij} = 1$) or not ($y_{ij} = 0$). This requires capturing abstract, non-literal semantics from visual input alone.

Our goal is to learn a powerful embedding function $g_\phi : \mathcal{X} \to \mathbb{R}^d$ that projects stickers into a semantic space where their pairwise cosine similarity, $\mathrm{sim}_\phi(s_i, s_j)$, directly reflects their semantic relationship. The final binary prediction for a pair is obtained by thresholding this similarity score.

### 3.2 DATASET CONSTRUCTION

To enable rigorous evaluation of sticker semantic similarity task, we construct the **S**ticker **S**emantic **S**imilarity (**Triple-S**) dataset. To the best of our knowledge, this is the first dataset specifically designed for pairwise sticker semantic similarity evaluation.

We use the 1,116 unique stickers from StickerQueries Chee et al. (2025b) to construct a sticker image-level semantic similarity dataset. StickerQueries is particularly suitable as it contains user search queries that aid in dataset construction. To the best of our knowledge, this is the first benchmark of its kind. However, naive construction based solely on these textual queries yields poor-quality semantic pairs, as the queries may not reliably reflect semantic intent and can contain indistinguishable irony or sarcasm so simple textual pairing is insufficient. To create a feasible and trustworthy benchmark, we incorporate human annotation.

**Annotation Process.** For each sticker $s_i$, we retrieve twenty candidate stickers, forming a candidate set $\rfloor_i^{(t)}$, where $t \in \{1, 2\}$ denotes the iteration. Annotators are asked to select stickers they consider semantically similar to $s_i$. To guide this process, annotators are encouraged to briefly reflect on the meaning of each sticker, but no strict constraints are imposed. If no semantically similar sticker is found, annotators may skip. Then, we denote the set of stickers selected by annotators as $\mathcal{L}_s \subseteq \mathcal{L}$.

The semantically similar set for sticker $s_i$ in iteration $t$ is then defined as:
$$\mathcal{S}_i^{(t)} = \rfloor_i^{(t)} \cup \{s_i\}.$$

Across the dataset, we have multiple such sets:
$$\mathcal{S} = \bigcup_{i=1}^n \bigcup_{t=1}^2 \mathcal{S}_i^{(t)},$$

where each $\mathcal{S}_i^{(t)} \in \mathcal{S}$ represents the semantically similar stickers for one anchor sticker in one iteration. In other words, each ground-truth sticker is represented in two independently constructed sets, capturing diverse semantic judgments and reducing annotation bias.

**Positive Pair Construction.** A pair of stickers $(s_i, s_j)$ is considered a *positive pair* ($y_{ij} = 1$) if they appear together in at least two semantically similar sets across the two iterations:
$$y_{ij} = 1 \quad \text{if} \quad \left|\{\mathcal{S}_k^{(t)} \in \mathcal{S} : s_i, s_j \in \mathcal{S}_k^{(t)}\}\right| \geq 2, \quad t \in \{1, 2\}.$$

In other words, both stickers must co-occur in multiple independently constructed sets to ensure consistent semantic similarity.

**Negative Pair Construction.** A pair $(s_i, s_j)$ is labeled as a *negative pair* ($y_{ij} = 0$) only if both of the following hold across the two iterations:

1. **Co-occurrence in candidate sets:** They appear together in candidate sets at least twice, but are never selected together in any semantically similar set:
$$\left|\{\rfloor_k^{(t)} : s_i, s_j \in \rfloor_k^{(t)}\}\right| \geq 2, \quad \text{and} \quad \left|\{\mathcal{S}_k^{(t)} : s_i, s_j \in \mathcal{S}_k^{(t)}\}\right| = 0, \quad t \in \{1, 2\}.$$

2. **Textual dissimilarity:** Let $Q_i$ and $Q_j$ denote the sets of textual queries associated with $s_i$ and $s_j$. We require both:

   - Low textual overlap: $|Q_i \cap Q_j| < 3$, and
   - Low semantic similarity: $\text{sim}_{\text{text}}(s_i, s_j) < 0.7$, where $\text{sim}_{\text{text}}$ is the cosine similarity of text embeddings.

This ensures that negative pairs are frequently co-present but semantically distinct, while the repeated annotation process improves reliability. All automatically extracted pairs were further validated manually, resulting in the first human-annotated dataset for sticker semantic similarity.

### 3.3 DATASET STATISTICS

The Triple-S benchmark consists of 905 semantic sticker pairs across 630 unique stickers, with 453 positive pairs (50.2%) and 449 negative pairs (49.6%), providing a balanced distribution. The annotations were collected from 49 participants who, on average, use stickers more than twice a day and are experienced sticker users, with diverse genders and ages. Table 2 summarizes the train, validation, and test splits.

Table 2: Statistics of the Triple-S benchmark dataset. The dataset contains 905 semantic sticker pairs across 630 unique stickers, with a roughly equal number of positive and negative pairs.

| Split | # pairs | # stickers | # positive pairs | # negative pairs |
|-------|---------|------------|------------------|------------------|
| Train | 765 | 477 | 394 (51.5%) | 371 (48.5%) |
| Test  | 140 | 153 | 62 (44.3%) | 78 (55.7%) |
| Total | 905 | 630 | 453 (50.2%) | 449 (49.6%) |

## 4 GSE: GENERAL STICKER ENCODER

Given the computational cost of LLMs and the lack of evaluation of general image encoders on sticker semantic similarity, we introduce, to our knowledge, **the first publicly available, General-purpose Sticker Encoder**.

### 4.1 GENERAL STICKER ENCODER CONSTRUCTION

Training a general sticker encoder requires substantial data, but the Triple-S dataset alone provides limited scale. To expand our training resources, we incorporate MultiChat Wang et al. (2025), where we use intention labels to indicate semantic similarity.

For each sticker pair $(s_i, s_j)$ in MultiChat, we assign a positive label ($y_{ij} = 1$) when both stickers share the same intention label, and a negative label ($y_{ij} = 0$) otherwise. This labeling approach generates a large collection of semantic sticker pairs suitable for training. We conduct manual reviews to ensure quality, and for evaluation, we use human-annotated pairs from StickerQueries to maintain benchmarking integrity. Then, we obtain 603,351 training pairs and 75,855 validation pairs. Combined with Triple-S benchmark dataset, the complete dataset contains 604,116 training pairs and 75,995 validation pairs.

We leverage the powerful pretrained image encoders by fine-tuning CLIP (Radford et al., 2021) on the combined dataset described above using a standard contrastive loss, aligning sticker images with their textual descriptions to learn a unified representation space without explicit pairwise similarity supervision. This allows the model to capture semantic relationships between visual and textual sticker content.

To evaluate the learned embeddings, we perform a binary semantic similarity task: for each test sticker pair $(s_i, s_j)$, we compute the cosine similarity $\text{sim}_\phi(s_i, s_j)$, and tune a threshold $\tau$ on a validation set to maximize F1, effectively forming a robust binary classifier. We refer to this trained model as the **General Sticker Encoder (GSE)**, which can generate semantic embeddings for any sticker and support downstream tasks such as similarity search, clustering, and recommendation.

## 4.2 POSSIBLE USAGES FOR ENCODER

There are several possible usage scenarios for the encoder; firstly, the sticker emotion classification task aims to categorize stickers into seven distinct emotion classes. Then, the sticker-to-sticker retrieval task aims to retrieve stickers that are semantically similar to a given query sticker.

## 5 TRIPLE-S BENCHMARK EXPERIMENTS

### 5.1 EXPERIMENT SETUP

The sticker semantic similarity task is formally defined in Section 3.1. Our experiment is designed to answer the following key research question (RQs):

**RQ1:** (**Baselines on Triple-S Benchmark**) To what extent can current image encoders perform the sticker semantic similarity task? How challenging is the Triple-S benchmark?

**Baselines and Implementation Details**   We benchmark current common methods on the Triple-S dataset:

- **CLIP** Radford et al. (2021): Robust zero-shot alignment capabilities

- **ViT** Dosovitskiy et al. (2020): Standard vision transformer architecture

- **DINOv2** Oquab et al. (2023): State-of-the-art self-supervised visual representations. For the above methods, we extract embeddings for each sticker, compute similarity scores from embedding pairs, and apply a threshold for classification.

- **ChatGLM-4V-Flash** GLM et al. (2024): Directly judges semantic relatedness of sticker pairs through prompting to generate similarity scores (see §A).

All experiments are conducted on a single NVIDIA A100 GPU. Evaluation is performed using Accuracy, Precision, Recall, F1, and ROC-AUC.

### 5.2 EXPERIMENT RESULTS

Table 3: Performance comparison on semantic similarity task. **Bold** indicates best performance, underline indicates second best. Acc.: Accuracy, Prec.: Precision, AUC: ROC AUC.

| | **Triple-S Benchmark** | | | | |
|---|---|---|---|---|---|
| **Model** | **Acc. ↑** | **AUC ↑** | **Recall ↑** | **F1 ↑** | **Prec. ↑** |
| CLIP | 0.439 | 0.476 | 1.000 | 0.610 | 0.439 |
| ViT | 0.439 | **0.617** | 1.000 | 0.610 | 0.439 |
| DINOv2 | 0.439 | 0.537 | 1.000 | 0.610 | 0.439 |
| ChatGLM-4V-Flash | **0.507** | 0.580 | 0.742 | 0.571 | **0.465** |

From Table 3, we observe that all standard image encoders (CLIP, ViT, DINOv2) produce nearly identical predictions across test pairs, resulting in the same Recall, F1, and Precision scores, which indicates they fail to capture the nuanced semantic differences between stickers. Although ChatGLM-4V-Flash achieves higher accuracy, its F1 score is lower than that of the image encoders, showing that stronger models also struggle with the dataset. Together, these results demonstrate that Triple-S is a challenging benchmark.

These observations highlight that relying on **superficial stylistic similarities or shallow semantic representations is insufficient for the sticker semantic similarity evaluation**. Consequently, there is a clear motivation for developing a General Sticker Encoder (GSE) capable of learning more robust semantic embeddings that better capture sticker-level nuances.

# 6 Effectiveness of General Sticker Encoder

## 6.1 Experiment Setup

### 6.1.1 Research Questions

We define the following research questions to evaluate the effectiveness of GSE:

**RQ2:** (**GSE Generalization Performance**) Does training on Triple-S increase generalizability on other unseen sticker semantic similarity datasets?

**RQ3:** (**GSE on Downstream Tasks**) Can GSE representations effectively transfer to sticker emotional classification and sticker-to-sticker retrieval tasks?

**RQ4:** (**GSE Ablation**) How does each dataset component contribute to GSE performance?

### 6.1.2 Dataset

**Sticker Semantic Similarity** In addition to the Triple-S benchmark dataset, we evaluate on the WXChallenge dataset WeChat (2023). We create sticker pairs from the WXChallenge dataset, which originally contains sticker-query pairs. Positive pairs are defined by exact query overlaps, while negative pairs are defined by low textual similarity (cosim $< 0.15$) computed using text2vec Ming (2022).

**Emotion Classification** SER30K Liu et al. (2022) contains 6,148 stickers in its test split, while MET-Meme Liaolian (2023) includes 3,994 memes. To evaluate on GSE, we randomly select one representative sticker from each emotion category as a reference. Each test sticker is then classified into the emotion category of the reference sticker with which it has the highest cosine similarity. Following prior work Chen et al. (2025), we restrict to the English subset of MET-Meme and report Accuracy, Precision, Recall, and F1.

**Sticker-to-Sticker Retrieval** We evaluate on WXChallenge and SER30K. On WXChallenge, stickers with the same query are aggregated, while on SER30K, aggregation is based on emotion categories. For each query, one sticker is randomly selected as the query image, and cosine similarity is computed across the corpus. Performance is reported using Recall@k for various $k$ values.

### 6.1.3 Baselines and Implementation Details

For emotion classification, we use the following baselines:

- **M3_Cat** Wang et al. (2024a): A multimodal matrix factorization model leveraging categorical information for sticker recommendation.

- **MGMCF** Zheng et al. (2025): Multi-Graph Multi-View Collaborative Filtering model capturing user-sticker interactions across different contexts.

- **MAM+Bert** Zhang et al. (2024): Attention-based multimodal model combining sticker image features with textual cues using BERT embeddings.

- **TGCA-PVT** Chen et al. (2024): Transformer-based graph cross-attention model with PVT backbone for multimodal sticker understanding.

- **MGHFT** Chen et al. (2025): Current state-of-the-art multimodal LLM for multi-view sticker interpretation.

- **MGHFT+GSE**: MGHFT with PVT backbone replaced by frozen GSE embeddings

For sticker-to-sticker retrieval, we use the same set of image encoder baselines as in the sticker semantic similarity task. The GSE is fine-tuned for 5 epochs with a learning rate of $1 \times 10^{-4}$ and batch size of 32. Hyperparameters are selected via grid search on a held-out validation set. All experiments are conducted on a single NVIDIA A100 GPU. For MGHFT-PVT+GSE, training is performed for 50 epochs following prior work.

Table 4: Zero-shot generalization performance on the unseen augmented WXChallenge dataset. The **Improv. (%)** row shows the percentage change of our model (GSE) over the second-best performer (underlined) for each metric. Acc.: Accuracy, Prec.: Precision, AUC: ROC AUC.

| | WXChallenge | | | | |
|---|---|---|---|---|---|
| **Model** | **Acc.** ↑ | **AUC** ↑ | **Recall** ↑ | **F1** ↑ | **Prec.** ↑ |
| CLIP | 0.607 | 0.651 | 0.668 | 0.496 | 0.394 |
| CLIP-CN | 0.543 | 0.699 | 0.814 | 0.508 | 0.369 |
| ViT | 0.325 | 0.556 | 0.971 | 0.454 | 0.297 |
| DINOv2 | 0.290 | 0.522 | **1.000** | 0.449 | 0.290 |
| **GSE (Ours)** | **0.665** (+9.6%) | **0.706** (+1.0%) | 0.642 | **0.526** (+3.5%) | **0.446** (+13.2%) |

## 6.2 GSE Generalization Performance (RQ2)

On unseen data, GSE achieves the highest scores across metrics—accuracy (0.665), ROC AUC (0.706), F1 (0.526), and precision (0.446)—a 9.6% relative improvement in accuracy over CLIP (0.607). Pretrained embeddings show a recall–precision trade-off: DINOv2 has perfect recall (1.000) but low precision (0.290), and ViT also has high recall (0.971) but poor precision (0.297). GSE balances recall (0.642) and precision (0.446), indicating better calibration for practical use.

These results demonstrate that GSE captures deeper semantic nuances in stickers compared to other baselines, making it a more reliable approach for evaluating sticker semantic similarity. Moreover, the effective generalization of GSE indicates that the Triple-S benchmark, combined with the additional datasets used for training, provides a robust foundation for learning a general-purpose sticker encoder capable of handling unseen data.

## 6.3 GSE on Downstream Tasks (RQ3) - Emotion Classification

Table 5: Performance comparison on the SER30K (top) and MET-Meme (bottom) datasets. Each table is split into models without training (left) and trained variants (right).

| SER30K – Not trained | | | SER30K – Trained | | |
|---|---|---|---|---|---|
| Model | Accuracy | F1 | Model | Accuracy | F1 |
| CLIP | 22.25% | 17.53% | MAM+Bert | 69.75% | 68.58% |
| ViT | 10.34% | 11.12% | TGCA-PVT | 71.63% | 70.93% |
| DINOv2 | 21.39% | 21.91% | MGHFT | 73.31% | 72.52% |
| **GSE (Ours)** | **31.69%** | **32.49%** | MGHFT+GSE | **74.27%** | **74.02%** |

| MET-Meme – Not trained | | | MET-Meme – Trained | | |
|---|---|---|---|---|---|
| Model | Accuracy | Precision | Recall | Model | Accuracy | Precision | Recall |
| CLIP | 17.88% | 15.74% | 14.46% | M3F_cat | 29.82% | 34.18% | 30.73% |
| ViT | 13.77% | 11.83% | 13.09% | MGMCF | 34.36% | **37.77%** | 34.38% |
| DINOv2 | 12.92% | 13.68% | 13.61% | MGHFT | 35.13% | 34.75% | 35.12% |
| **GSE (Ours)** | **18.91%** | **16.16%** | **14.57%** | MGHFT+GSE | **38.17%** | 37.53% | **38.17%** |

As shown in Table 5, the features from GSE provide a significant performance boost. When used as an image encoder, **GSE outperforms all generic image encoders (CLIP, ViT, DINOv2)** on both the SER30K and MET-Meme datasets. For instance, on SER30K, GSE achieves an Accuracy of **31.69%** and an F1 of **32.49%**, a substantial improvement over the best pretrained baseline (DINOv2 at 21.39% and 21.91%). Most notably, when we replace the vision backbone of the state-of-the-art (SOTA) MGHFT model with our GSE embeddings, the resulting model **MGHFT+GSE sets a new SOTA**. It achieves the highest accuracy and F1 metrics on SER30K and a consistent **+3%** absolute improvement across all metrics on MET-Meme. This demonstrates that our **GSE embeddings capture rich, complementary semantic information that is directly transferable and highly effective for affective tasks**, even surpassing features from models specifically trained for emotion recognition.

## 6.4 GSE on Downstream Tasks (RQ3) - Sticker-to-Sticker Retrieval

Table 6: Sticker-to-Sticker Retrieval Performance on SER30K and WXChallenge datasets.

| Img Encoder | SER30K (Recall@K) | | | | WXChallenge (Recall@K) | | | |
|---|---|---|---|---|---|---|---|---|
| | @5 | @10 | @20 | @100 | @5 | @10 | @20 | @100 |
| CLIP | 0.012 | 0.017 | 0.036 | 0.132 | 0.371 | 0.343 | 0.336 | 0.290 |
| ViT | 0.009 | 0.012 | 0.023 | 0.094 | 0.257 | 0.243 | 0.264 | 0.234 |
| DINOv2 | 0.009 | 0.012 | 0.026 | 0.098 | 0.400 | 0.343 | 0.307 | 0.293 |
| **GSE (Ours)** | **0.012** | **0.019** | **0.045** | **0.167** | **0.543** | **0.529** | **0.464** | **0.419** |
| Improv. | +0.05% | +0.21% | +0.86% | +3.55% | +46.36% | +54.23% | +38.10% | +44.48% |

Our proposed **GSE method demonstrates superior performance in sticker-to-sticker retrieval** across both datasets. On SER30K, GSE consistently outperforms baseline methods, with gains increasing at higher recall levels (up to +3.55% at Recall@100). On WXChallenge, GSE shows substantial improvements across all recall levels, ranging from +38.08% to +54.23%. These results indicate that GSE maintains stronger retrieval precision even with larger candidate pools. These findings further highlight the versatility and generalizability of GSE trained on the Triple-S dataset. In **addition to semantic similarity, GSE also performs effectively on related tasks such as sticker emotion classification and sticker-to-sticker retrieval**, demonstrating its broad applicability.

## 6.5 GSE Ablation (RQ4)

Table 7: Ablation Study of GSE Components on WXChallenge and MET-Meme datasets.

| Component | WXChallenge | | | | MET-Meme | | | |
|---|---|---|---|---|---|---|---|---|
| | Acc. | F1 | Prec. | ROC AUC | Acc. | F1 | Prec. | Recall |
| CLIP Baseline | 60.66 | 49.58 | 39.42 | 65.05 | 17.88 | 15.13 | 15.74 | 14.46 |
| Trained on MultiChat | 60.94 | 49.39 | 39.53 | 66.34 | 18.16 | 14.68 | 15.45 | 14.63 |
| Trained on Triple-S | 56.60 | 51.06 | 37.91 | 68.85 | 18.86 | 16.59 | 17.00 | 14.40 |
| **GSE (Ours)** | **66.51** | **52.60** | **44.57** | **70.61** | **18.91** | **17.92** | **16.16** | **14.57** |
| Improvement | +5.85 | +1.54 | +5.04 | +1.76 | +1.03 | +2.79 | +0.42 | +0.11 |

The ablation study demonstrates the contribution of individual components in GSE. The final GSE achieves the best performance across all metrics, with significant improvements in Accuracy (+5.85%) and Precision (+5.04%) over the CLIP baseline. GSE shows consistent improvements, particularly in F1 score (+2.79%), indicating better performance across all classes. Training on individual datasets (Triple-S or MultiChat) shows mixed results, but the combined approach in **GSE yields the most robust performance across both datasets**. The improvements are more substantial on WXChallenge, suggesting our method better handles the complexity of sticker similarity tasks compared to meme understanding in MET-Meme.

## 7 Conclusion and Future Work

In this work, we make several key contributions. First, we formally define the **Sticker Semantic Similarity** task. Second, we introduce the **Triple-S** benchmark, the first benchmark for the visual sticker semantic similarity task, consisting of 905 human-annotated positive and negative sticker pairs. Third, we benchmark common methods on Triple-S, revealing their limitations. Finally, we present the **General Sticker Encoder (GSE), a lightweight and versatile sticker encoder that provides strong performance on unseen data and generalizes effectively**. Benchmarking on Triple-S demonstrates that **standard image encoders are insufficient for reliably assessing sticker semantic similarity** or semantic preservation. In contrast, GSE consistently captures deeper semantic nuances and performs effectively on related tasks such as emotion classification and sticker-to-sticker retrieval. In future work, we plan to extend the Triple-S benchmark to additional scenarios and incorporate emerging models, further advancing research on sticker semantic understanding and evaluation.

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

## A  APPENDIX

The LLM evaluation prompt is structured as follows:

> **Instruction for VLM to obtain sticker semantics**
>
> *You are a sticker similarity judge. Compare these two stickers and evaluate: 1. Semantic similarity score (0-1, where 1=identical meaning) 2. Brief reasoning for your score. Return in JSON format: {"score": 0.xx, "reason": "..."}*

