# OpenReview forum: "Triple-S: A Sticker Semantic Similarity Benchmark with General Sticker Encoder"
_ICLR.cc/2026/Conference — Submitted to ICLR 2026_

### Official Review · Reviewer_WS3h · 2025-10-29

**Soundness:** 3
**Presentation:** 2
**Contribution:** 2
**Rating:** 4
**Confidence:** 2

**Summary:**

This paper introduces the task of sticker semantic similarity examining the challenge of encoding the diverse and symbolic semantics of digital stickers. The main contribution is Triple-S, the first human-annotated benchmark dataset designed specifically for sticker semantic similarity evaluation, with 905 positive and negative sticker pairs. The authors demonstrate that recent vision and general multimodal models mainly attempt to capture the nuanced meaning in stickers. For examining this deficiency, the authors proposes the general sticker encoder for improving the semantic representation and fusion capabilities for this specific visual-symbolic modality.

**Strengths:**

S1) The definition of sticker semantic similarity addresses notable terms of multimodal communication research, moving beyond literal image understanding to symbolic, expressive content.

S2)  The proposed benchmark can be one of the contributions for future research in this unique domain.

S3)  The work effectively validates the limitations of existing foundation models, demonstrating that their pre-training on natural images or general vision-language data does not translate well to stylized, cartoon-like, and symbolic sticker semantics.

**Weaknesses:**

W1) A size of 905 annotated pairs is small and limited, raising notable concerns about the potential for overfitting and the generalization capability.

W2) The novelty in the multimodal fusion architecture of the General Sticker Encoder needs clearer articulation. What is new? There is no specific difference between the findings of the current study and implications of prior research.

W2) Sticker semantics are heavily context- and culture-dependent.

**Questions:**

Q1) Could the authors elaborate on the specific architectural choices in the general sticker encoder for allowing to capture symbolic semantics?

Q2) What is the performance drop when depending on  the visual encoder features versus the fused representation?

Q3) Could the authors test any other methods (e.g. self-supervised learning) for addressing this task?

---

### Official Review · Reviewer_Zc1T · 2025-11-01

**Soundness:** 2
**Presentation:** 2
**Contribution:** 2
**Rating:** 2
**Confidence:** 5

**Summary:**

This work focuses on the sticker semantic similarity task. The authors manually created a benchmark, Triple-S, which consists of 905 sample pairs, and trained a lightweight General Sticker Encoder (GSE) using the training set from the benchmark. Experimental results on both Triple-S and other test datasets show that GSE outperforms CLIP, ViT, and DINOv2.

**Strengths:**

1. The paper constructs a high-quality sticker similarity benchmark through manual annotation, which is meaningful for the relevant community.

2. The experimental design is comprehensive, demonstrating the effectiveness of the collected data through ablation studies and showcasing the performance advantages of GSE through experiments on multiple test datasets.

3. The overall structure of the paper is clear and easy to understand.

**Weaknesses:**

1. The scale of the constructed benchmark is relatively small, with only 140 sample pairs used for testing. I am concerned whether such a small-scale test set is sufficient to reflect the model's performance.

2. The methods compared are limited. Most experiments only compare CLIP, DINOv2, and ViT. More advanced CLIP-based methods (e.g., EVA-CLIP, SigLIP) or MLLM-based methods (e.g., UniME-V2, VLM2Vec-V2) need to be compared.

3. The technical contribution of the GSE design is limited, as it only fine-tunes CLIP using the constructed dataset.

**Questions:**

1. The writing logic could be further improved:
    a. In the second paragraph, the phrase "In addition to existing baselines, providing a feasible evaluation method xxx" is unclear. The terms "baselines" and "evaluation method" are not well-defined, which may cause confusion.
    b. The second paragraph mentions that manually annotated datasets suffer from issues like subjectivity and labor-intensiveness, but the dataset constructed by the authors is also manually annotated, so the issues of subjectivity and labor-intensiveness have not been addressed.

2. What are the differences between GSE and StickerCLIP in terms of methodology?

3. Why is the performance of GSE not reported in Table 3?

---

### Official Review · Reviewer_yX6H · 2025-11-01

**Soundness:** 2
**Presentation:** 3
**Contribution:** 2
**Rating:** 2
**Confidence:** 4

**Summary:**

This paper's main goal is to address the challenge of understanding nuanced sticker semantics. To do this, the authors first introduce Triple-S, a new, human-annotated benchmark of 905 sticker pairs, and demonstrate that standard encoders (e.g., CLIP and ViT) and a VLM (ChatGLM-4V-Flash) struggle to do well on it. They then propose the General Sticker Encoder (GSE), a CLIP image encoder fine-tuned on their Triple-S data combined with the MultiChat dataset. The paper evaluates GSE on a range of downstream tasks, showing it outperforms baselines on unseen sticker similarity and retrieval datasets (i.e., WXChallenge, SER30K) and achieves new state-of-the-art performance in emotion classification when its embeddings are integrated into an existing multimodal model.

**Strengths:**

- The paper formally provides a carefully curated benchmark (Triple-S) designed to test the nuanced understanding of stickers, and benchmarks common vision encoders (CLIP, ViT, DINOv2), effectively demonstrating their limitations in capturing nuanced sticker semantics and justifying the need for a better encoder.

- The proposed General Sticker Encoder (GSE) demonstrates effective transfer learning and outperforms other popular vision encoders. When integrated into a multimodal LLM, it achieves new state-of-the-art results on downstream emotion classification tasks, validating that the embeddings are robust and generalizable

**Weaknesses:**

- The paper only benchmarks one VLM (ChatGLM-4V-Flash) in Section 5. Given the test set's tiny size (140 pairs), there should be results for stronger commercial or larger open-source VLMs. The paper fails to reveal the state of the art on this task.

- The ablation study (Table 7) shows that training on Triple-S or MultiChat alone gives marginal benefit, but training on both provides a huge boost. There is no analysis or reasoning on why the small-scale training set of Triple-S can bring that much improvement when combined with MultiChat. Furthermore, there should also be discussions to rule out any potential data leakage between its Triple-S training set and the WXChallenge test set.

**Questions:**

Please see the weaknesses section.

---

### Author Response · Authors · 2025-11-13
**Our Sincere Thanks for Your Thoughtful Review**

Dear Reviewers,

We would like to express our heartfelt gratitude for the time, care, and thought you devoted to reviewing our paper. Your insightful and constructive comments have given us valuable perspectives and ideas for improvement, and we are truly grateful for the effort you put into helping our work grow stronger.

We will take your feedback to heart as we continue developing and refining our research. Thank you once again for your generosity and dedication to advancing the field — your feedback has been both encouraging and deeply meaningful to us.

With sincere appreciation,
The Authors

---

### Meta-Review · Area_Chair_1URZ · 2026-01-09

**Summary:**

This paper introduces Triple-S, a benchmark for sticker semantic similarity, together with a lightweight General Sticker Encoder (GSE). The work is motivated by the observation that existing general-purpose vision and multimodal encoders struggle to capture the nuanced and symbolic semantics of stickers. Reviewers generally agree that the task formulation is reasonable and that the authors make a genuine effort to construct a human-annotated, fine-grained pairwise benchmark. The release of Triple-S as a standardized evaluation resource is viewed as a positive contribution to an underexplored area,However, the overall assessment is that the paper’s technical contribution is limited. While the benchmark itself is useful, the scale of the dataset is relatively small, and the proposed GSE model largely follows standard fine-tuning and contrastive learning practices. Reviewers were not convinced that the modeling or experimental results provide sufficiently strong new insights beyond what is already achievable with existing encoders and data. As a result, despite a clear motivation and solid engineering effort, the paper does not reach the level of novelty and depth expected for acceptance.

**Reviewer Concerns:**

Concerns that were partially addressed:
1. The authors clarified the annotation pipeline and pair construction strategy, which improves confidence in the quality of the Triple-S benchmark.
2. The paper more clearly positioned sticker semantic similarity as distinct from sentiment classification or retrieval-based tasks.
Concerns that remain outstanding:
1. Limited dataset scale and scope: Although carefully curated, Triple-S contains fewer than 1,000 labeled pairs. Reviewers questioned whether this scale is sufficient to support strong conclusions about general sticker semantics or to justify training a “general” encoder.
2. Incremental modeling contribution: GSE is essentially a fine-tuned CLIP-style encoder trained with additional sticker-related data. The architecture and learning strategy are standard, and the paper does not introduce new representation learning techniques specific to stickers.
3. Evaluation interpretation: While improvements over baselines are reported, many gains are modest or rely on downstream task setups that may favor the proposed encoder. Reviewers expressed concern that some comparisons conflate representation quality with task-specific training choices.
4. Broader impact and generalization: It remains unclear how well the benchmark and encoder would transfer to more diverse sticker styles, cultures, or unseen platforms beyond the datasets considered.
Overall, reviewers view the benchmark as a useful starting point, but the accompanying model and experimental evidence are not yet strong enough to justify acceptance.

**Reviewer Scores:**

Reviewer yX6H:
Reviewer yX6H raised concrete and substantive concerns regarding the limited evaluation against strong VLM baselines, the small test set size, and the unclear source of gains when combining Triple-S with MultiChat. While the discussion may help clarify some experimental details and reduce concerns about potential data leakage, these issues are central to the reviewer’s assessment of soundness and contribution. It is therefore unlikely that this reviewer would substantially revise their score. At most, the score might be slightly softened in confidence, but would remain in the reject range.
Reviewer Zc1T:
Reviewer Zc1T expressed strong concerns about dataset scale, limited baseline coverage, and the incremental nature of the proposed General Sticker Encoder. These concerns are consistent, clearly articulated, and aligned with the reviewer’s high confidence rating. Even with further discussion, it is unlikely that these fundamental concerns about novelty and evaluation sufficiency would be resolved. As a result, the reviewer’s score would almost certainly remain unchanged.
Reviewer WS3h:
Reviewer WS3h acknowledged the relevance of the task and the potential value of the benchmark, and already placed the paper marginally below the acceptance threshold. With fuller participation in the discussion, this reviewer might be partially reassured by clarifications regarding benchmark construction and the motivation for focusing on lightweight encoders. As a result, this reviewer could plausibly move slightly upward (e.g., from “marginally below acceptance” to a weak reject or borderline score), but would still likely stop short of a clear accept due to unresolved concerns about dataset scale and methodological novelty.

---

### Decision · Program_Chairs · 2026-01-26

Reject